# Nonlinear Elastic Wave Energy Imaging for the Detection and Localization of In-Sight and Out-of-Sight Defects in Composites

**Joost Segers [1,\*], Saeid Hedayatrasa [1,2], Gaétan Poelman [1], Wim Van Paepegem [1] and Mathias Kersemans [1]**

1   Mechanics of Materials and Structures (UGent-MMS), Department of Materials, Textiles and Chemical Engineering (MaTCh), Ghent University, Technologiepark-Zwijnaarde 46, 9052 Zwijnaarde, Belgium; Saeid.Hedayatrasa@UGent.be (S.H.); Gaetan.Poelman@UGent.be (P.G.); Wim.VanPaepegem@UGent.be (V.P.W.); Mathias.Kersemans@UGent.be (K.M.)
2   SIM Program M3 DETECT-IV, Technologiepark-Zwijnaarde 48, B-9052 Zwijnaarde, Belgium
*   Correspondence: Joost.Segers@UGent.be

**Abstract:** In this study, both linear and nonlinear vibrational defect imaging is performed for a cross-ply carbon fiber-reinforced polymer (CFRP) plate with artificial delaminations and for a quasi-isotropic CFRP with delaminations at the edge. The measured broadband chirp vibrational response is decomposed into different components: the linear response and the nonlinear response in terms of the higher harmonics. This decomposition is performed using the short-time Fourier transformation combined with bandpass filtering in the time-frequency domain. The linear and nonlinear vibrational response of the defect is analyzed by calculation of the defect-to-background ratio. Damage maps are created using band power calculation, which does not require any user-input nor prior information about the inspected sample. It is shown that the damage map resulting from the linear band power shows high sensitivity to shallow defects, while the damage map associated to the nonlinear band power shows a high sensitivity to both shallow and deep defects. Finally, a baseline-free framework is proposed for the detection and localization of out-of-sight damage. The damage is localized by source localization of the observed nonlinear wave components in the wavenumber domain.

**Keywords:** composites; NDT; local defect resonance; nonlinearity; laser Doppler vibrometry; band power; short-time Fourier transform; out-of-sight damage detection

## 1. Introduction

Composite materials such as carbon fiber-reinforced polymers (CFRP) are currently used in a wide range of industries. Because of their design flexibility and high specific stiffness and strength, these composite materials are often used in load-bearing components where they replace the traditionally used metals. However, a concern in the use of composites is that their layered structure is susceptible to internal damage which can be introduced during the manufacturing process as well as the operational life. One example is low-velocity impact damage which is typically referred to as barely visible impact damage because the impact introduces delaminations in between the layers while the impacted surface remains (almost) intact. Internal damage leads to a strong local decrease of the strength (and stiffness) of the component which can eventually result in unexpected failure.

In this study, a baseline-free vibrometric non-destructive testing (NDT) approach is proposed for localizing shallow and deep internal damage in CFRP components. The method uses low-power (<10 W) piezoelectric excitation combined with non-contact scanning laser Doppler vibrometer (SLDV)

measurements. Using SLDV measurements, it was shown earlier that defected plates can show local resonances at the location of the defects [1–3]. The frequencies at which these local resonances occur are referred to as 'local defect resonance (LDR) frequencies'. Depending on the size, shape and depth of the defects, these LDR frequencies are typically in the 1 to 100 kHz range. LDR behavior has been investigated for different defect types: flat bottom holes [3–5], artificial delaminations [4,5] (e.g., Teflon delaminations), disbonds [6] and barely visible impact damage [3,7]. If these defects are relatively shallow (depth up to 50% of the sample's thickness), they can be detected by a manual or automated [8] search for LDR behavior in the broadband response of the component. However, if the defects are located deeper than the mid-thickness of the component, no local resonances occur as the reduction in the local bending stiffness at the defect becomes too limited [9].

Apart from local defect resonance, other wave–defect interactions can be exploited for NDT using full wavefield measurements. For instance, the defect-related shift in local wavenumber is exploited in local wavenumber estimation methods [10–13]. Alternatively, the typical increase in vibrational amplitude at damage is exploited using the (weighted) root mean square energy calculation [14]. An increased sensitivity to damage is found when performing wavenumber filtering prior to the calculation of the energy maps [15,16]. In this way, both the shift in wavenumber as well as the shift in energy are exploited. These vibrometric methods show promising results for NDT but also have their limits in terms of minimal defect size and maximum defect depth [17,18].

In order to further increase the sensitivity to small and deep damage, nonlinear defect imaging techniques were developed, such as nonlinear elastic wave spectroscopy (NEWS) [19–25] and nonlinear elastic wave modulation spectroscopy (NEWMS) [26–29]. These methods focus on the detection of nonlinear frequency components, for example higher harmonics, in the output signal which are then correlated to defects. The nonlinear response of the defect is caused by classical material nonlinearity combined with multiple complex contact mechanisms which are triggered at relatively high vibrational amplitudes [30]. In order to achieve the required level of vibrational amplitude at the defect when using low-power piezoelectric actuators, the defect can be excited at LDR frequency [31–33]. The current authors showed recently that the nonlinear LDR response of a deep backside-delamination can be observed experimentally [34].

When using nonlinear defect imaging techniques, the nonlinear components have to be extracted from the output response of the sample. If a sine excitation or a narrowband chirp excitation is used, the nonlinear higher harmonic components can easily be differentiated using standard fast Fourier transformation (FFT). However, in the case of a broadband chirp excitation, this is no longer possible (see also explanation in Section 3). One possible solution is to use a combination of inverse filtering and phase symmetry analysis for extraction of the second and third higher harmonic components [35,36]. Here, a novel method is proposed using short-time Fourier transformation (STFT) and time-frequency bandpass filtering. This proposed method does not require multiple phase coded excitations and advanced data handling which is needed in the case of phase symmetry analysis and inverse filtering. Also it allows harmonic components of any specified order to be extracted.

The different steps of the damage detection procedure discussed in this paper are schematically shown in Figure 1. For each step, the corresponding section number is indicated.

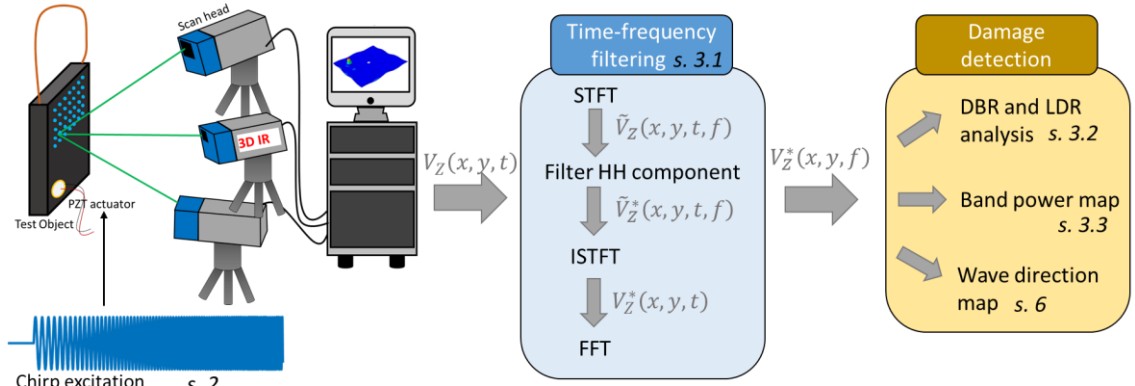

**Figure 1.** Schematic overview of all steps in the damage detection procedure.

## 2. Materials and Measurements

Two CFRP test specimens were investigated for damage (see Figure 2).

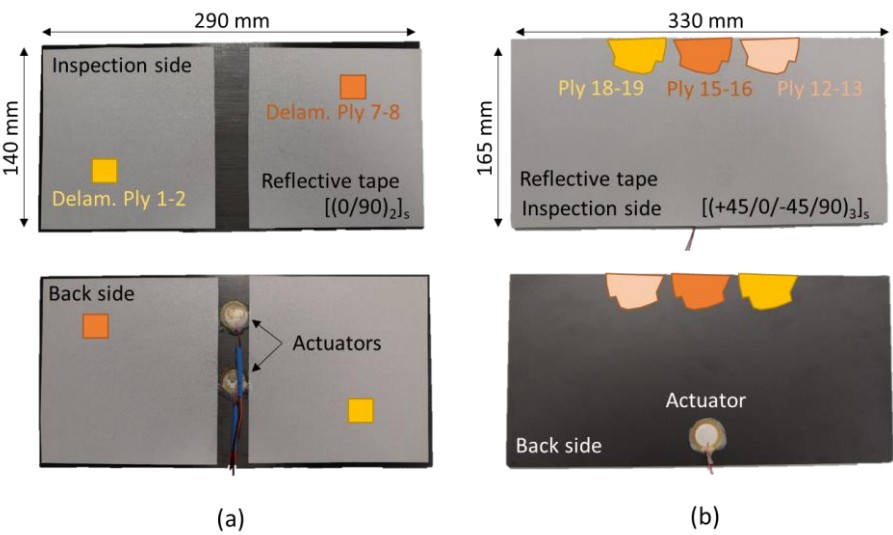

**Figure 2.** Carbon fiber-reinforced polymer (CFRP) components with defects. (**a**) Cross-ply $[(0/90)_2]_s$ plate with artificial shallow (ply 1–ply 2) and backside (ply 7–8) delamination. (**b**) Quasi-isotropic $[(+45/0/-45/90)_3]_s$ plate with side-delaminations at the top side.

The first plate (Figure 2a) measured $290 \times 140 \times 2.1$ mm$^3$ and was manufactured by unidirectional CFRP prepreg according to cross-ply layup $[(0/90)_2]_s$. When stacking the prepreg, two artificial delaminations were introduced using $20 \times 20$ mm$^2$ inserts made out of a double layer of 25 μm thin brass foil. One insert was placed between the first and second ply (shallow defect), while the other insert was placed between the seventh and eighth ply (deep backside defect). The exact shape and size of the induced delamination was not fully controllable, because the epoxy resin tended to flow partially in-between the insert's layers during the autoclave curing cycle.

The second test specimen (see Figure 2b) was a $330 \times 165 \times 5.5$ mm$^3$ CFRP plate with quasi-isotropic layup $[(+45/0/-45/90)_3]_s$. Side-delaminations were created at the top side of the plate using a fine razor blade. The exact spatial dimensions of the induced side-delaminations were unknown. The numbers of the plies in between which the blade was inserted are indicated on the figure. Note that the majority of the damage is located relatively deep into the 24 ply plate. This type of side-delaminations is representative of damage created by improper cutting and (rivet hole) drilling of multi-layer CFRP plates.

The components were excited using piezoelectric patches (type EPZ-20MS64W and EPZ-27MS44W from Ekulit, with a diameter of 12 mm and 20 mm respectively). The patches were bonded to the CFRP components (see Figure 2) using phenyl salicylate. As there was no prior knowledge on the damage and its corresponding LDR frequencies, a broadband chirp excitation was required to make sure that the LDRs were excited. In that way, it was assured that the nonlinear behavior of a defect will be triggered in an efficient way.

The first coupon with delaminations was excited using a chirp signal with linearly increasing frequency from 10 to 60 kHz and length of 36 ms. The chirp signal was zero-padded at the start and at the end for 2 ms resulting in a total duration of 40 ms. The voltage of excitation was amplified to 100 $V_{pp}$ using a Falco System WMA-300 voltage amplifier. The coupon with side-delaminations was excited with a 100 $V_{pp}$ chirp of 16 ms and frequency from 5 to 100 kHz. The electrical power delivered by the amplifier to the piezoelectric patches was only around 5 Watts.

The response of the CFRP specimens was recorded at a uniformly spaced grid of scan points with grid spacing 2 mm using a 3D infrared SLDV (Polytec PSV-500 3D Xtra). The sampling frequency was set to 0.625 MS/s. Only the out-of-plane velocity component was discussed in this study. However, note that similar observations can be made by analyzing the in-plane velocity component [3].

Although a SLDV with infrared lasers (wavelength λ = 1550 nm), showing high sensitivity on even black surfaces was employed, the signal-to-noise ratio (SNR) was further improved by additional measures. The inspected surfaces were covered with reflective tape (3M™ Scotchlite™ 580E-10). Using reflective tape, the resolution of the out-of-plane velocity component was around 0.15 $\frac{\mu m}{s}$ / $\sqrt{Hz}$. In addition, for the component with side-delaminations, 3 averages were made to further increase the SNR. These SNR enhancing measures were favorable as the amplitude of the nonlinear components in the output response were several orders of magnitude smaller compared to the linear components.

## 3. Data Processing

### 3.1. Time-Frequency Filtering

As explained in the previous section, a broadband excitation signal was used to make sure that potential LDRs were excited. The broadband nature of the excitation signal made higher harmonics extraction using the classical fast Fourier transform (FFT) impossible. This is graphically illustrated in Figure 3 for the CFRP plate with artificial delaminations. The component was excited using a burst chirp signal of duration 40 ms and frequency range 10 kHz to 60 kHz. For instance, at the time instance $t$ = 6 ms, the instantaneous chirp excitation frequency was 15 kHz. This resulted in a linear response of the sample at $f_{lin}$ = 15 kHz combined with the potential presence of nonlinear higher harmonic components: second harmonic at $f_{HH2}$ = 30 kHz, third harmonic at $f_{HH3}$ = 45 kHz, etc. (see Figure 3b,c, indicated in red). These higher harmonic components were of low amplitude and as such they were overshadowed by the linear part of the output response at 16 ms (i.e., $f_{lin}$ = 30 kHz, indicated in green) and 28 ms (i.e., $f_{lin}$ = 45 kHz) respectively. This is further indicated as 'overlap' on Figure 3c. Note again that reducing the frequency bandwidth of excitation was not an option as it would reduce the likelihood of exciting LDRs. As an alternative, a novel post-processing approach using short-time Fourier transform (STFT) and bandpass filtering in the time-frequency domain was proposed.

The decomposition of the output response into its linear and nonlinear components using bandpass filtering in the time-frequency domain is graphically illustrated in Figure 4.

First, the measured velocity response $V_Z(x, y, t)$ is transformed from the time domain to the time-frequency domain using STFT:

$$\widetilde{V}_Z(x, y, t(l), f(k)) = \frac{1}{M} \sum_{m=1}^{M} V_Z^l(x, y, t(m)) \, e^{-2\pi i \frac{mk}{M}}$$
$$\text{with } V_Z^l(x, y, t(m)) = V_Z(x, y, t(m + lH)) \, w(m) \tag{1}$$

where $L$ is the number of time-divisions and each time-division is represented by index $l = 1, 2 \ldots L$. $M$ is the length of each time-division expressed in number of samples, while $m = 1, 2 \ldots M$ is the local time index within the time-division $l$. $H$ is the hop size or distance between successive time-divisions, expressed in number of samples. The array $w(m)$ is a window which is multiplied with each time-division $l$ to avoid spectral leakage. The output after STFT is $\widetilde{V}_Z(x, y, t(l), f(k))$ in which $k = 1, 2 \ldots K$ is the frequency index and $K = \frac{M}{2}$ (Nyquist). The '~' indicate that the signal is expressed in the time-frequency domain. The STFT is performed in Matlab using the implementation according to reference [37], with hop size $H = 66$, window length $M = 512$ and number of time divisions $L = 1 + \frac{N-M}{H} = 372$ (with $N = 25\,000$, total number of time samples). A Hanning window is employed for $w(m)$.

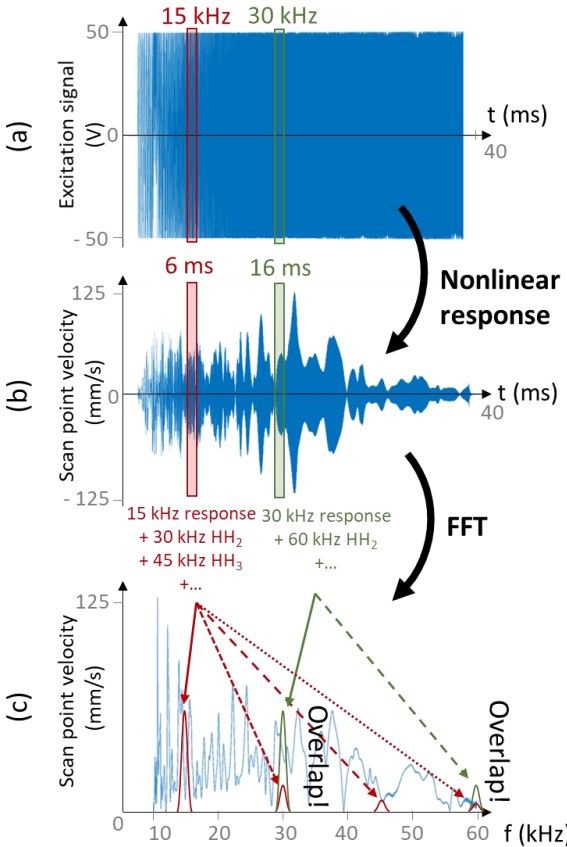

**Figure 3.** (**a**) Burst chirp excitation signal in time domain, (**b**,**c**) velocity response in time and frequency domain (using fast Fourier transformation (FFT)), respectively.

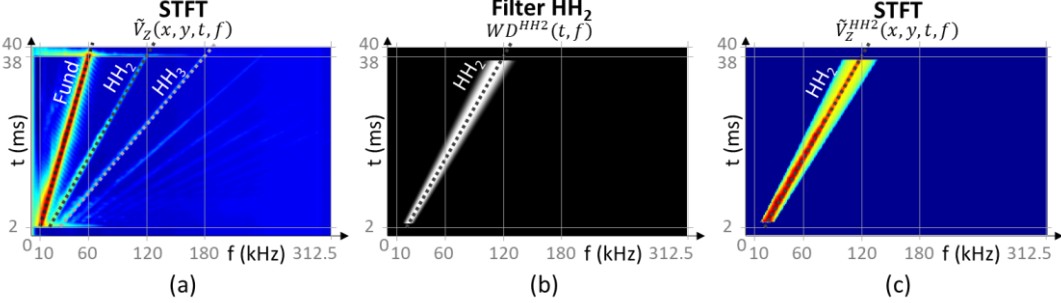

**Figure 4.** Extraction of the second higher harmonic component *HH2* for the test specimen with artificial delaminations: (**a**) Average short-time Fourier transform of the measured out-of-plane velocity response, (**b**) Bandpass filter around the *HH2* curve, (**c**) Extracted *HH2* component in time-frequency domain.

The absolute value of $\widetilde{V}_Z(x,y,t,f)$, averaged over all scan points is shown in Figure 4a for the coupon with artificial delaminations. The linear response of the component is visible as a line of increased amplitude from $f_{start}$ = 10 kHz to $f_{end}$ = 60 kHz. Next to this linear response, multiple lines are found corresponding to the higher harmonic components. The second and third higher harmonic components are marked $HH_2$ and $HH_3$.

In the next step, a time-frequency bandpass filter $WD^*(t,f)$ is constructed around one of the components of interest. The asterisk * refers to the filtered component: *lin*, *HH2* or *HH3*. The filter is constructed as:

$$WD^*(t,f) = WD_1^*(t,f).WD_2(t)$$
with :
$$WD_1^*(t,f)$$
$$= \begin{cases} 1 & \left|f^* - f_{lin}(t)\right| < \frac{FT(t)}{2} \\ 0 & \left|f^* - f_{lin}(t)\right| > \frac{FT(t)}{2} + \frac{\pi}{4}BW(t) \\ \frac{1}{2} + \frac{1}{2}cos\left(\frac{4\left(|f^* - f_{lin}(t)| - \frac{FT(t)}{2}\right)}{BW(t)}\right) & elsewhere \end{cases}$$
$$WD_2(t) = \begin{cases} 0 & t\langle 0.06\ t_{meas}\ \text{OR}\ t\rangle 0.94\ t_{meas} \\ 1 & elsewhere \end{cases}$$
and
$$f_{lin}(t) = f_{start} + k(t)\ (f_{end} - f_{start})$$
$$FT(t) = FT_1 + k(t)\ (FT_2 - FT_1)$$
$$BW(t) = BW_1 + k(t)\ (BW_2 - BW_1)$$
$$f^* = f\ \text{for linear component extraction}$$
$$f^* = f/2\ \text{for sec ond HH component extraction}$$
$$f^* = f/3\ \text{for third HH component extraction}$$
$$k(t) = \frac{t - 0.05\ t_{meas}}{0.90\ t_{meas}}$$

(2)

where $f_{start}$ and $f_{end}$ are the start and end frequency of the chirp, respectively. $t_{meas}$ is the total measurement time, $k(t)$ represents the chirp advance and $f_{lin}(t)$ is the instantaneous chirp excitation frequency at time $t$. The filter is Tukey shaped with time-dependent flat top length $FT(t)$ and edge bandwidth $BW(t)$. The term $WD_2(t)$ limits the filter in time domain such that the response is only retained when the excitation signal is active. This is advised because the abrupt start and end of the burst chirp excitation signal, at 2 ms and 38 ms respectively, results in a broadband response of the sample (see Figure 4a).

As an example, Figure 4b shows the filter for extracting the second harmonic component $WD^{HH2}$ (with $f^* = 2f$). The excitation signal properties are: $f_{start}$ = 10 kHz, $f_{end}$ = 60 kHz and $t_{meas}$ = 40 ms and the filter settings are: flat top length $FT_1$ = 3 kHz (at start), $FT_2$ = 7 kHz (at end) and edge band width $BW_1$ = 3 kHz (at start), $BW_2$ = 7 kHz (at end). These filter settings are chosen such that the window fully envelops only the *HH2* component (see Figure 4c).

As a final step, inverse STFT is performed to transform the bandpass filtered signal from the time-frequency domain back to the time domain:

$$V_Z^{l*}(x,y,t(m)) = \sum_{k=1}^{K} \widetilde{V}_Z(x,y,t(l),f(k))\ WD^*(t(l),f(k))\ e^{2\pi i\frac{m\,k}{K}}$$
$$\rightarrow V_Z^*(x,y,t(n)) = \frac{H}{E_{wv}} \sum_{l=1}^{L} V_Z^{l*}(x,y,t(n-lH))\ w(n-lH)\ v(n-lH)$$

(3)

Note that the time divisions $V_Z^{l*}$ are shifted in time ($m = n - lH$) and added in order to obtain the final filtered signal $V_Z^*(x,y,t(n))$. The function $v$ is a window which helps to fulfill the correct overlap-and-add (OLA) conditions: $\sum_{l=1}^{L} w(n-lH)\ v(n-lH)$ must be constant over time index $n$ and

$E_{wv} = \sum\limits_{m=1}^{M} w(m) \, v(m)$. Here, a Hanning window is used and the OLA requirement was verified using the "*OLAExam*" software [37].

It is often desired to process the obtained velocity component $V_Z^*(x, y, t)$ in the frequency domain instead of the time domain. Therefore, FFT is performed:

$$V_Z^*(x, y, f(k)) = \frac{1}{N} \sum_{n=1}^{N} V_Z^*(x, y, t(n)) \, e^{-2\pi i \frac{nk}{N}} \tag{4}$$

### 3.2. Defect-to-Background Ratio

The defect-to-background ratio *DBR* is introduced in order to quantify the increase in amplitude at the defect relative to the surrounding damage-free material. The *DBR* is calculated at each frequency $f$:

$$DBR^*(f) = \frac{\Omega_{healthy}}{\Omega_{defect}} \frac{\sum_{i=1}^{n_{defect}} V_Z^*(x_i, y_i, f)}{\sum_{i=1}^{n_{healthy}} V_Z^*(x_i, y_i, f)} \tag{5}$$

where $\Omega_{defect}$ is the known defected area that contains $n_{defect}$ measurement points and $\Omega_{healthy}$ is the surrounding healthy area with $n_{healthy}$ measurement points. $V_Z^*(x_i, y_i, f)$ is the frequency-specific velocity amplitude at location $(x_i, y_i)$, which may correspond to any of the filtered components discussed in the previous section, for instance the second harmonic component $V_Z^* = V_Z^{HH2}$. Thus, the *DBR* equals the average amplitude of the (filtered) vibration velocity at the defect's location compared to the average amplitude of the (filtered) vibration velocity at the remainder of the coupon.

### 3.3. Band Power Caculation

The band power ($BP^*$) represents the broadband vibrational energy of the sample corresponding to velocity component $V_Z^*$. The $BP^*$ is defined as [9]:

$$BP^*(x, y, f(k_1), f(k_2)) = \frac{1}{k_2 - k_1} \sum_{k=k_1}^{k_2} V_Z^*(x, y, f(k))^2 \tag{6}$$

The $BP^*$ is calculated between two frequencies: $f(k_1)$ and $f(k_2)$ with $1 \le k_1 < k_2 \le \frac{N}{2}$ (Nyquist). Note again that the $BP^*$ can be calculated for each filtered velocity component (as represented by the asterisk *). For instance, the band power of the second harmonic ($V_Z^{HH2}$) is denoted as $BP^{HH2}$. The BP as calculated using Equation (6) gives the vibrational power over a certain frequency band. A damaged area is typically characterized by a local reduction in bending stiffness combined with nonlinear contact conditions. As a result, an increased fundamental and higher harmonic vibrational amplitude could be expected (especially at an LDR frequency). This would result in an increased $BP^{lin}$, $BP^{HH2}$ and $BP^{HH3}$ value at the location of damage.

## 4. Detection of Artificial Delaminations

In this section, the measurement results of the CFRP plate with one shallow and one deep backside delamination (see Figure 2a) are analyzed. The filtering process explained in Section 3.1 is used to obtain the linear response as well as the second and third higher harmonic components. First, the *DBR* ratio is calculated to evaluate the linear and nonlinear response of the defect at different frequencies. Using the *DBR* curves, the local defect resonance behavior is discussed. Next, the band power maps are calculated and their sensitivity to the defects is evaluated.

### 4.1. Defect-to-Background Ratio and Local Defect Resonance

Using Equation (5), the $DBR^{lin}(f)$, $DBR^{HH2}(f)$ and $DBR^{HH3}(f)$ curves are calculated for both the shallow as well as the deep delamination. The curves are shown in Figure 5a,e, respectively. In order to improve the readability of the graphs, the frequency axis is scaled with the order of the higher harmonic. As an example, the red line in Figure 5a indicates the *DBR* of the linear component at 17.3 kHz, the second harmonic component at $2 \times 17.3 = 34.6$ kHz and the third harmonic component at $3 \times 17.3 = 51.9$ kHz.

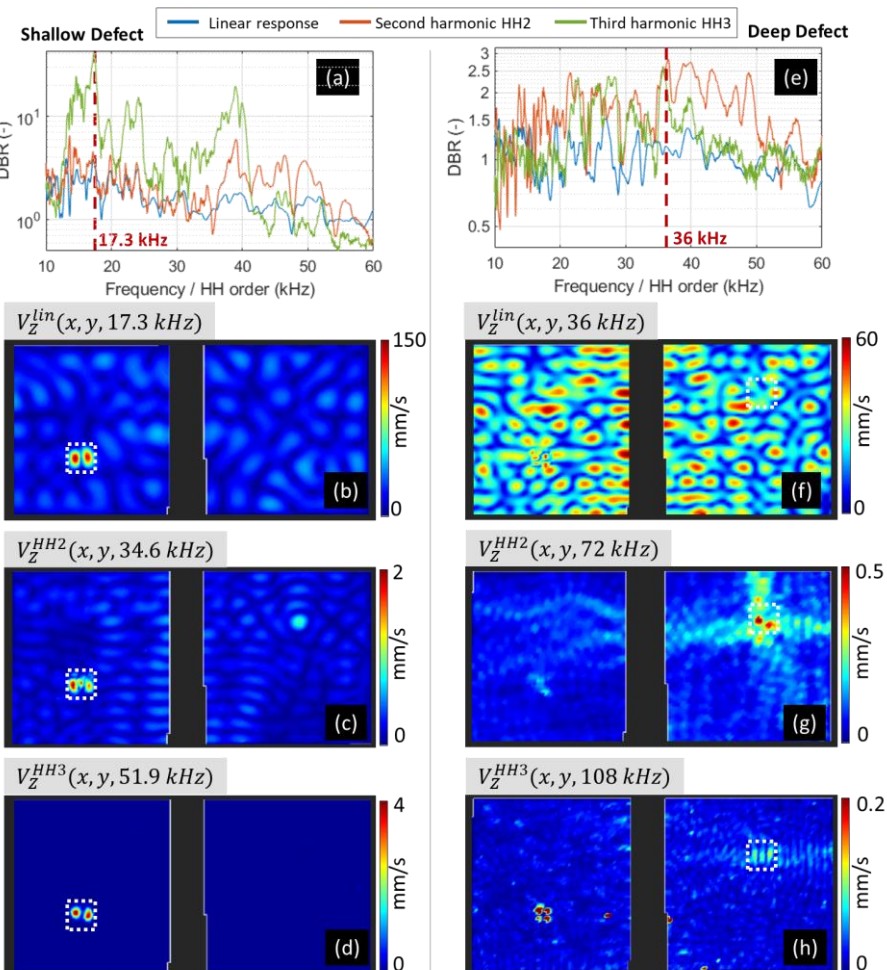

**Figure 5.** Defect-to-background ratio (*DBR*) curves for linear and nonlinear higher harmonic components with amplitude maps at local maxima. Results for (**a-d**) shallow delamination, and (**e-h**) backside-delamination.

The shallow delamination shows pronounced LDR behavior due to the high reduction in local bending stiffness (see also [9]). This LDR behavior corresponds to the local maxima in the $DBR^{lin}(f)$ curve, for instance at 17.3 kHz. The operational deflection shape corresponding to the linear component at 17.3 kHz reveals the (second order) LDR of the delamination, see Figure 5b.

The *DBR* curves of the higher harmonics show a value higher than 1 at the majority of frequency bins. This indicates that the delaminations show distinct nonlinear behavior. As expected, the higher harmonics are observed, especially when the defect is under LDR behavior. This is seen in Figure 5a because the $DBR^{HH2}(f)$ and $DBR^{HH3}(f)$ curves show local maxima at the same scaled frequency as where the $DBR^{lin}(f)$ is high. As an example the amplitude maps are shown for the second and third higher harmonic corresponding to the LDR frequency of 17.3 kHz (see Figure 5c,d). An intense nonlinear response of the defect is revealed by these amplitude maps. Especially for the third harmonic,

the defect is revealed with much greater contrast compared to the linear defect response. As such, examining these higher harmonic components for LDR behavior leads to an improved sensitivity to defects compared to the examination of the linear response.

For the deep delamination near the backside, no LDR behavior is observed in the linear response due to the limited reduction in local bending stiffness at the delamination [9]. Indeed, the $DBR^{lin}(f)$ curve (see blue curve in Figure 5e) is close to 1 at all frequencies and does not show any significant local maximum.

On the other hand, the *DBR* curves corresponding to the higher harmonic components do reveal an increased nonlinear activity at the defect at a multitude of frequencies. As an example, the vibration maps of the linear and higher harmonic components are shown for the frequency of 36 kHz. No sign of the delamination is visible in the linear response at 36 kHz (Figure 5f). However, an increased activity at the defect is seen in both the second and the third higher harmonic components (Figure 5g,h, respectively). The efficient generation of nonlinear components at the deep defect is related to LDR of the thin material section between the delamination and the backside [34].

These observations indicate that the monitoring of higher harmonic components leads to a more robust defect detection and localization compared to the monitoring of only the linear response. Moreover, it is shown that a delamination which is close to the backside can exclusively be detected through the analysis of the higher harmonic components.

### 4.2. Band Power

The band power is calculated over the total excitation bandwidth according to Equation (6) and is shown in Figure 6. The band power of the linear component $BP^{lin}$ reveals only the shallow delamination (see Figure 6a). There is no sign of the backside-delamination because this area does not show an increase in the linear velocity component. This observation is in agreement with the *DBR* curves in Figure 5a,e: the shallow delamination has $DBR^{lin}(f) > 1$ for the majority of the frequency bins, while the deep backside-delamination has $DBR^{lin}(f) \approx 1$ for most frequency bins.

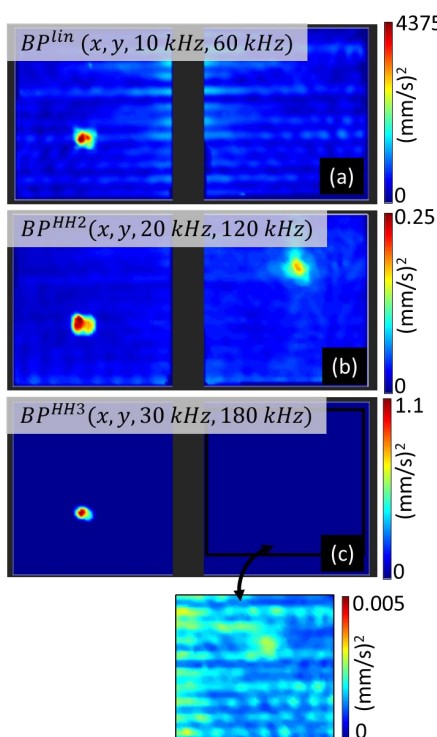

**Figure 6.** Band power maps calculated using Equation (6) for (**a**) linear velocity component, (**b**) second harmonic component and (**c**) third harmonic component.

The band power map of the second harmonic component ($BP^{HH2}$) reveals the presence of both delaminations (see Figure 6b). As such, this band power map can be used as a robust indicator for detection of both shallow and deep defects. The third harmonic band power map reveals the shallow delamination with very high contrast (see Figure 6c). However, for the deep defect, the third harmonic components are not efficiently transferred through the thickness of the component. This makes the backside-delamination hardly distinguishable from the damage-free material in the $BP^{HH3}$ map. The exact origin of this absent *HH3* is currently being investigated.

## 5. Detection of Side-Delaminations

In this section, the nonlinear response of side-delaminations (see Figure 2b) is investigated.

Time-frequency filtering (see Section 3.1) is used to extract the linear and the second higher harmonic responses from the measurement data. Next, the *DBR* curves are calculated (see Equation (5) for these two filtered signals. Both curves are shown in Figure 7a. All side-delaminations are located at a depth equal to or bigger than half the thickness of the CFRP sample. As a result, no LDR behavior is detected in the linear velocity component (see Figure 7b,c) and the $DBR^{lin}$ curve fluctuates around 1. Also, in the corresponding band power map (see Figure 7d), the absence of an increased linear response is observed. As such, this linear component is not efficient for the detection of these relatively deep defects.

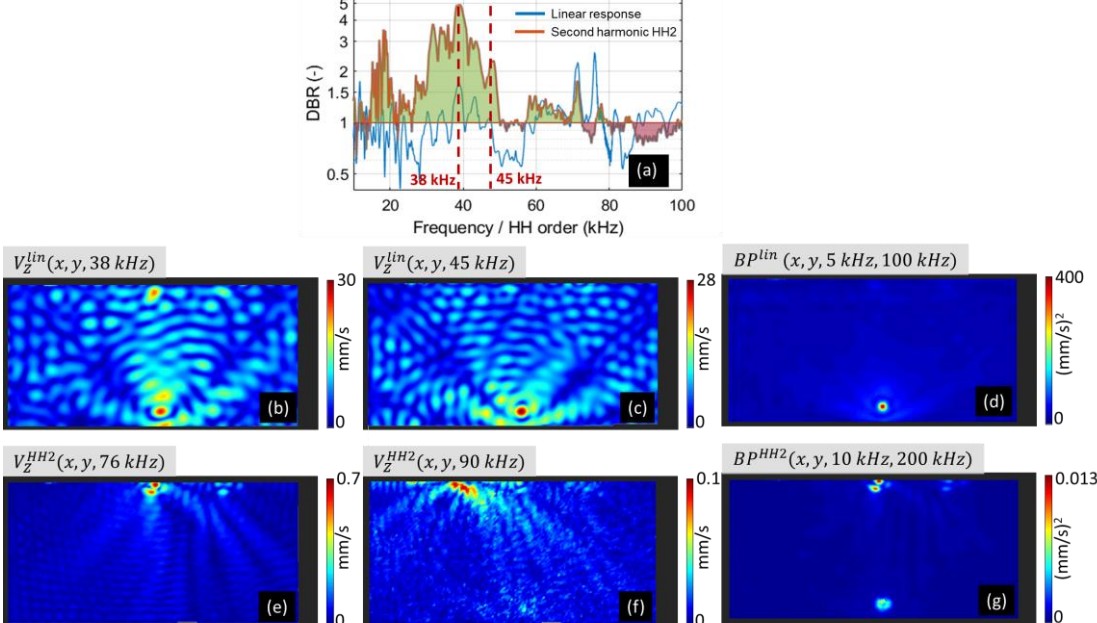

**Figure 7.** Side-delamination response analysis: (**a**) defect-to-background ratio of linear and second harmonic component, (**b-d**) linear response amplitude at 38 kHz, at 45 kHz and band power map, (**e-g**) second harmonic response amplitude at 2 × 38 kHz, at 2 × 45 kHz and band power map.

The *DBR* curve corresponding to the second harmonic component reveals an increased nonlinear activity at the defect. The maximum *DBR* is reached at 76 kHz (i.e., 2 × 38 kHz) and the corresponding amplitude map (Figure 7e) shows the nonlinear source behavior of the delaminated edge. While at 76 kHz the main nonlinear response is found at the middle of the top edge, other frequencies reveal other parts of the damage. As an additional example, the amplitude map of *HH2* at 2 × 45 kHz = 90 kHz (Figure 7f) reveals the nonlinear behavior of the deeper side-delamination more towards the left of the top edge.

The *DBR* curve of the second harmonic component (see Figure 7a) indicates that the defect behaves as a source of second harmonics for the majority of the frequencies. To make this more clear, the area

beneath the curve is colored green when there is an increased second harmonic component at the defect compared to the damage-free material, i.e., $DBR^{HH2}(f) > 1$, and it is colored red when $DBR^{HH2}(f) < 1$. This observation is exploited in the calculation of the nonlinear band power map. Indeed, a strong increase in the band power is observed for the second harmonic component (see Figure 7g). The band power map is dominated by the nonlinear source behavior of the side-delamination at the middle of the top edge. Note that the band power of the second harmonic also indicates that the excitation PZT itself induces a limited amount of nonlinearity in the system.

The linear and nonlinear response of the side-delaminations match well with the response of the artificial backside-delamination discussed in the previous section. In both cases, defect detection is possible by searching for an increased higher harmonic activity at specific frequencies at which LDR is induced, or simply by calculating the nonlinear band power over the total excitation bandwidth.

## 6. Opportunities for Out-of-Sight Damage Detection

In Sections 4.1 and 5, it was observed that higher harmonic frequency components are generated at the location of a defect, especially when the defect is locally resonating. These higher harmonic components can radiate away from the defect into the damage-free material. This is clearly visible in the amplitude map of the second harmonic component shown in Figure 7e,f. The intensity of the harmonic component decreases with the distance to the side-delamination because of geometrical wave spreading (~distance) and wave attenuation (which is relatively high in this frequency range).

The nonlinear radiation of defects is further exploited in order to detect out-of-sight defects. Such defects are typically encountered if certain parts of the component are hidden and cannot be reached by the laser beams, or simply if one wants to reduce the measurement time. As an example, the inspection area of the CFRP plate with side-delaminations is reduced to two small square areas of around $40 \times 40$ mm$^2$ (indicated *Area 1* and *Area 2* in Figure 8a). The measurement results at these small areas are used to not only detect [32], but also to localize the out-of-sight side-delaminations.

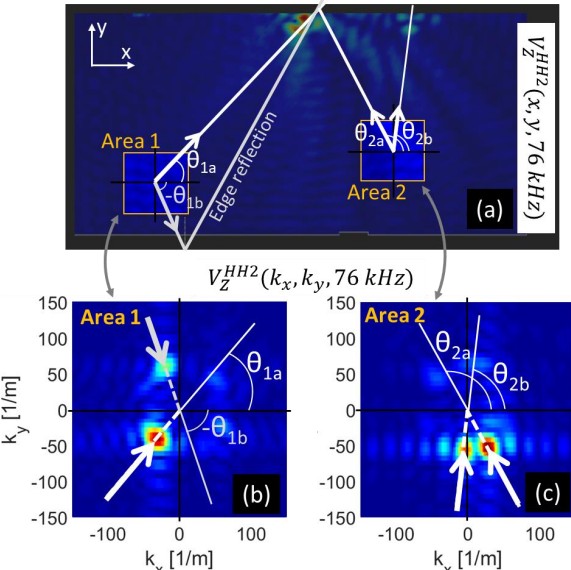

**Figure 8.** Detection of out-of-sight side-damage by using the second harmonic component measured at two small areas: (**a**) second harmonic amplitude map at 76 kHz with indication of the measurement areas, (**b,c**) wavenumber maps corresponding to the second harmonic component at 76 kHz with indication of the direction of the incoming wavefield.

To start, the second harmonic component is extracted using time-frequency filtering (see Section 3.1). The subsequent processing steps are applied individually at each scan area. First, the extracted second

harmonic component is transformed from the time domain to the wavenumber-frequency domain using three-dimensional FFT:

$$V_z^{HH2}\left(k_x(r), k_y(s), f(k)\right)$$
$$= \frac{1}{N.P.Q} \sum_{p=1}^{P} \left( \sum_{q=1}^{Q} \left( \sum_{n=1}^{N} V_Z^{HH2}(x(p), y(q), t(n)) e^{-2\pi i \frac{k\,n}{N}} \right) e^{-2\pi i \frac{s\,q}{Q}} \right) e^{-2\pi i \frac{r\,p}{P}} \tag{7}$$

where the number of scan points in horizontal and vertical directions are denoted as $P$ and $Q$, respectively. In order to 'increase' the resolution in wavenumber domain, the $V_Z^{HH2}$ signal is zero-padded in $x$ and $y$ direction. The resulting signal $V_z^{HH2}\left(k_x(r), k_y(s), f(k)\right)$ is now represented in the wavenumber-frequency domain where $k_x$ and $k_y$ are the wavenumbers in horizontal and vertical direction, respectively.

The wavenumber maps corresponding to the nonlinear *HH2* response at $f$ = 76 kHz, $V_z^{HH2}\left(k_x, k_y, 76kHz\right)$ are shown in Figure 8b,c. Distinctive spots of high intensity are found in both wavenumber maps. These spots correspond to the second harmonic radiation induced by a nonlinear source (defect). The direction of the source can be found as: $\theta = \arctan\left(\frac{k_y}{k_x}\right)$ if $k_x < 0$ or $\theta = \arctan\left(\frac{k_y}{k_x}\right) + \pi$ if $k_x > 0$. In Figure 8, arrows are drawn in these directions. The arrows correctly point towards the source of the higher harmonic components, namely the side-delaminations at the top side of the component. Note that for one of the observed directions in Figure 8b, the wavefield originates at the side-delamination but is reflected at the bottom edge. The edge is perceived as a virtual nonlinear source, and as such can be employed to further improve the localization of the actual defect. Hence, the out-of-sight damage can be detected and even localized without the need for a baseline measurement.

One important requirement for out-of-sight defect detection is that the defect must behave as a source of nonlinear components. At 76 kHz, this was the case for the delamination at the center and the delamination at the right side of the top edge. In order to detect other defects, other frequencies have to be evaluated. For instance at 90 kHz, the delamination at the left is detected as shown by Figure 9.

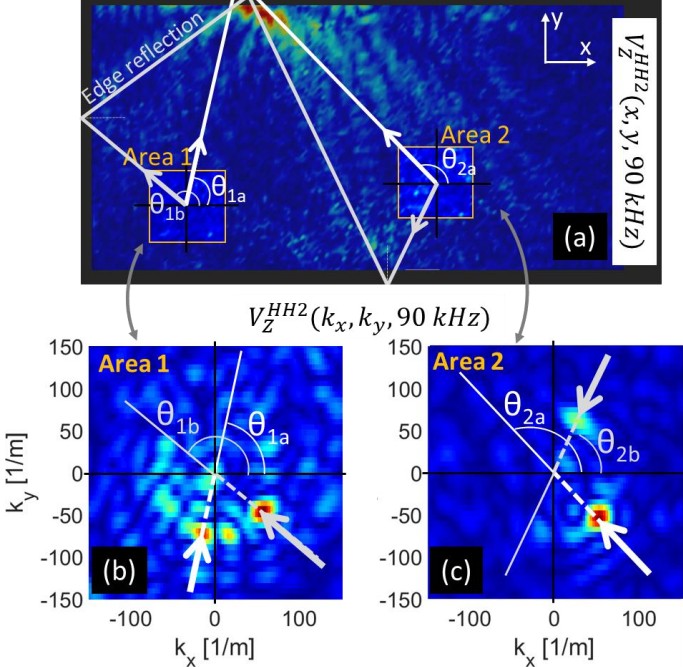

**Figure 9.** Detection of out-of-sight side-damage by using the second harmonic component measured at two small areas: (**a**) second harmonic amplitude map at 90 kHz with indication of the measurement areas, (**b**,**c**) wavenumber maps corresponding to the second harmonic component at 90 kHz with indication of the direction of the incoming wavefield.

## 7. Conclusions

The linear and nonlinear vibrational response of defects in CFRP coupons is investigated. The investigation is performed using experimental data obtained from two CFRP components: (i) a cross-ply CFRP plate with both a shallow and a deep artificial delamination; and (ii) a quasi-isotropic CFRP plate with side-delaminations. Broadband vibrations are introduced using low-power piezoelectric actuators while the full field velocity response is recorded with an infrared scanning laser Doppler vibrometer.

Time-frequency filtering, relying on (inverse) short-time Fourier transformations, is proposed to extracted linear and nonlinear velocity components (i.e., higher harmonics) out of the out-of-plane velocity measurement signal. This procedure enables the separation of the different harmonic components within a broadband response signal.

The linear and the nonlinear vibrational components are analyzed in a function of frequency by calculation of the defect-to-background ratio. For the shallow delamination, the linear velocity component shows local defect resonance behavior at specific frequencies. The high vibrational amplitude at local defect resonance results in the efficient generation of nonlinear higher harmonic components. For the deep artificial delamination and for the side-delaminations, no local defect resonance behavior is observed in the linear velocity component. These defects are located relatively close to the backside which causes the local defect resonances to be present only at the (invisible) backside. However, the nonlinear vibrational components, generated by the defect, radiate through the thickness and can be detected from the inspection side.

Broadband band power maps for both linear and nonlinear response are introduced. These broadband band power maps significantly improve the defect detectability. The nonlinear broadband band power maps, in particular, provide a high-contrast imaging of shallow as well as deep damage.

Finally, the developed procedures are applied for baseline-free detection and localization of out-of-sight damage. This is illustrated for the detection of side-delaminations. The vibrations are measured only at a few small areas of the component. For each scan area, the direction of the waves are determined. The derived directions point correctly to the side-delaminations.

**Author Contributions:** Conceptualization, J.S., M.K. and S.H.; methodology, J.S.; software, J.S.; validation, S.H.; resources, M.K. and W.V.P.; writing—original draft preparation, J.S.; writing—review and editing, S.H., G.P., M.K. and W.V.P.; supervision, M.K. and W.V.P.; funding acquisition, M.K. and W.V.P. All authors have read and agreed to the published version of the manuscript.

**Funding:** The authors acknowledge the funding of through Fonds voor Wetenschappelijk Onderzoek FWO (fellowships 1148018N, 1S11520N and 12T5418N) and the SBO project DETECT-IV (Grant no. 160455), which fits in the SIM research program MacroModelMat (M3) coordinated by Siemens (Siemens Digital Industries Software, Belgium) and funded by SIM (Strategic Initiative Materials in Flanders) and VLAIO (Flemish government agency Flanders Innovation and Entrepreneurship).

**Conflicts of Interest:** The authors declare no conflict of interest.

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
