# Peer review of "Nonlinear Elastic Wave Energy Imaging for the Detection and Localization of In-Sight and Out-of-Sight Defects in Composites"

_applsci, doi:10.3390/app10113924_

Round 1

Reviewer 1 Report

Several comments here for the authors to improve their manuscript:

  1. What's the sensitivity, or resolution of SLDV used in the study?
  2. The authors suggest to add a systematic measurement setup for the study. 
  3. Why did the authors use two different kinds of piezoelectric patches? What's the resonant frequencies of them?
  4. What did the authors mean the piezoelectric patches use ± 5 Watt of power?
  5. Why the piezoelectric patches placed like the figure 1 shown? 

Reviewer 2 Report

Good work, although I see numerous practical hurdles to be overcome before it develops into a practical inspection technique there is great potential for the approach. Not sure why you refer to "damages" instead of "damage" in the early stages of the paper.

Reviewer 3 Report

The authors study vibrational defect imaging of CFRP plates. They perform Fourier transforms and bandpass filtering to determine the linear and nonlinear vibrational response of defects. The paper is an interesting piece of work, well written and easy to understand. The authors supply sufficient experimental as well as theoretical data to substantiate their results. I see nothing in the paper that required amendment. The manuscript can be published as is.

Reviewer 4 Report

In this study, artificial delaminations embedded in cross-ply carbon fiber reinforced polymer (CFRP) plate is determined by using the short-time-Fourier transformation combined with bandpass filtering in the time-frequency domain. Although the topic and analyzed tools are not novel, the paper combined well and clear presents on the research-field outcome. In reviewer's opinions, the paper can be accepted before minor revision. Some drawbacks can be improvement in this manuscript.

1) In text, the word "manuscript" should be revised as "paper" or "study". After publication, this paper is formally academic achievement. It has to show the specialty.

2) In the last paragraph on Section I. Introduction, from line 84-92, the description is redundant. It can be deleted to save the pages.

3) In Figure 2(c), the diminishing waveform (red line) is hard to understand for what meant the authors presented. They are not overlapping with the peak or dip of blue lines or green lines.

4) The conclusion is too long to miss the focal contribution in this paper.

Round 2

Reviewer 1 Report

The authors have addressed all the comments.